# Lon1 Inactivation Downregulates Autophagic Flux and Brassinosteroid Biogenesis, Modulating Mitochondrial Proportion and Seed Development in *Arabidopsis*

**DOI:** 10.3390/ijms25105425

**Published:** 2024-05-16

**Authors:** Ce Song, Yuqi Hou, Tiantian Li, Yinyin Liu, Xian-Ao Wang, Wumei Qu, Lei Li

**Affiliations:** Frontiers Science Center for Cell Responses, Department of Plant Biology and Ecology, College of Life Sciences, Nankai University, Tianjin 300071, China; 1120170355@mail.nankai.edu.cn (C.S.); 1120200476@mail.nankai.edu.cn (Y.H.); 1120230701@mail.nankai.edu.cn (T.L.); 2120211144@mail.nankai.edu.cn (Y.L.); 2120221508@mail.nankai.edu.cn (X.-A.W.); 2120231446@mail.nankai.edu.cn (W.Q.)

**Keywords:** *Arabidopsis*, mitochondrion, Lon1, autophagy, brassinosteroid

## Abstract

Mitochondrial protein homeostasis is crucially regulated by protein degradation processes involving both mitochondrial proteases and cytosolic autophagy. However, it remains unclear how plant cells regulate autophagy in the scenario of lacking a major mitochondrial Lon1 protease. In this study, we observed a notable downregulation of core autophagy proteins in *Arabidopsis* Lon1 knockout mutant *lon1-1* and *lon1-2*, supporting the alterations in the relative proportions of mitochondrial and vacuolar proteins over total proteins in the plant cells. To delve deeper into understanding the roles of the mitochondrial protease Lon1 and autophagy in maintaining mitochondrial protein homeostasis and plant development, we generated the *lon1-2atg5-1* double mutant by incorporating the loss-of-function mutation of the autophagy core protein ATG5, known as *atg5-1*. The double mutant exhibited a blend of phenotypes, characterized by short plants and early senescence, mirroring those observed in the individual single mutants. Accordingly, distinct transcriptome alterations were evident in each of the single mutants, while the double mutant displayed a unique amalgamation of transcriptional responses. Heightened severity, particularly evident in reduced seed numbers and abnormal embryo development, was observed in the double mutant. Notably, aberrations in protein storage vacuoles (PSVs) and oil bodies were evident in the single and double mutants. Gene Ontology (GO) and Kyoto Encyclopedia of Genes and Genomes (KEGG) analyses of genes concurrently downregulated in *lon1-2*, *atg5-1*, and *lon1-2atg5-1* unveiled a significant suppression of genes associated with brassinosteroid (BR) biosynthesis and homeostasis. This downregulation likely contributes to the observed abnormalities in seed and embryo development in the mutants.

## 1. Introduction

Despite its counterintuitive nature, the breakdown of cellular components is crucial for the survival of all living organisms. Protein degradation plays a vital role in clearing malfunctioning or unnecessary proteins and regulating developmental and stress response processes [1]. Plant endosymbiotic organelle mitochondria contain approximately 2000–2500 proteins, tightly regulated through protein synthesis and degradation mechanisms. The degradation of mitochondrial proteins in plants involves mitochondrial-localized proteases, cytosolic autophagic-associated vacuolar degradation, and the ubiquitin–proteasomal degradation pathway.

The Lon protease is an ATP-dependent serine protease found in all living organisms, named after the long filament phenotype of *Escherichia coli* mutant cells [2]. In bacteria and non-plant organisms, the Lon protease is involved in protein quality control by recognizing unstructured protein segments or degradation tags and degrading unfolded and misfolded proteins [3]. In land plants, Lon proteases were found to degrade damaged or misfolded proteins in peroxisomes and mitochondria. The Lon protease forms homo-oligomers and has multiple domains, including a substrate binding domain, an ATPase domain, and a serine-type proteolytic domain [4,5]. Despite the similarities between plant and non-plant Lon proteases, the mechanisms of substrate recognition and oligomerization in plant organelles are not yet fully understood [3]. The evolution of a small Lon gene family in land plants suggests a functional diversification of Lon proteases in these organisms [6]. The *Arabidopsis* genome contains four Lon proteins, with Lon1 being well studied for its role in proteostasis mainly in mitochondria. In *lon1* mutants, the degradation rates of more than 200 mitochondrial proteins, including ribosomal proteins, electron transport chain subunits, and tricarboxylic acid cycle enzymes, were significantly altered compared to wild type [7]. A member of the Lon protease family, LON2, an ATP-dependent serine protease and chaperone, is crucial for maintaining plant peroxisomes and regulating their degradation through selective autophagy known as pexophagy [8]. However, it remains unclear how Lon1 interacts with autophagic degradation.

Autophagy degrades and recycles cellular material and nutrients through the use of lysosomes in mammalian cells and vacuoles in yeasts and plants. It is a widespread process in eukaryotic organisms. Autophagy core proteins can generally be divided into four functional complexes, including the ATG1/ATG13 kinase complex, the PI3K (phosphatidylinositol 3-kinase) complex, the ATG9 complex, and the ATG8/ATG12 ubiquitin-like conjugation system [9]. They all participate in regulating the induction of plant cell autophagy, the formation of autophagosomes, transport, and fusion with vacuoles, among other processes [9,10]. Autophagosomes engulf specific organelles for selective degradation. Mitochondria are a well-defined target for selective autophagy in yeast and mammalian cells [11,12]. The recognition of mitochondria for degradation through selective autophagy, known as mitophagy, has been well understood in yeast and animal model systems, as the protein regulators targeting mitochondria for autophagic degradation have been extensively characterized [13]. ATG11 was found to play an essential role in plant mitochondria degradation [14]. Most recently, plant mitophagy was found to be activated by carbon starvation, senescence, and specific mitochondrial stresses but not nitrogen starvation and general stresses [15]. A few potential mitophagy receptors and regulatory proteins involving FMT (friendly), TraB-family proteins, and FLZs (FCS-like zinc finger) were also characterized [16,17,18,19].

Recently, we previously reported that Lon1 dysfunction leads to unfolded protein responses (UPRs) characterized by enhanced ethylene production in *Arabidopsis* [20]. In this study, we reanalyzed the transcriptome changes in *lon1* mutants and discovered a general downregulation of autophagy core proteins, including ATG8 isoforms, at the transcript level. We further validated this downregulation at the protein level, which leads to a reduced autophagy flux, potentially contributing to the alteration of the relative proportion of mitochondrial and vacuolar proteins in plant cells. Moreover, *lon1* mutants were crossed with an autophagy-deficient mutant, *atg5-1*, to investigate the contribution of two independent degradation pathways to phenotype formation and transcript-level responses. The double mutant *lon1-2atg5-1* exhibited a combined phenotype from the single mutants, supported by the combined transcript-level changes. While distinct changes at the transcript level were apparent in the single mutants, a common downregulation of genes involved in brassinosteroid (BR) biosynthesis and homeostasis was observed in both single and double mutants. This supports the common abnormal seed and embryo development phenotype in single and double mutants. Taken together, this study, along with previous findings, reveals that Lon1 dysfunction upregulates the unfolded protein response but downregulates autophagic flux, thereby increasing the relative proportion of mitochondrial proteins to maintain steady respiration. The downregulation of BR biosynthesis and homeostasis is a common feature of dysfunctional Lon1 and autophagy, possibly associated with a shared mediator that deserves further investigations.

## 2. Results

### 2.1. Autophagy Flux Is Lowered in lon1 through the Downregulation of Autophagy Core Proteins at the Level of Transcripts

Autophagy was reported to maintain basal-level mitochondrial protein homeostasis and mitochondria clearance under mitochondrial stresses or over developmental transitions [15,17,19,21]. This raises the question of how autophagy is modulated in *lon1* mutant lines. Surprisingly, a reanalysis of reported RNA deep sequencing data [20] found that core autophagy genes were significant downregulated in *lon1-2* (Figure 1A). *ATG2*, *ATG8s* (*ATG8A*, *E*, *F*, *G*, and *H*), *VTI12*, and *CFS1* proteins are involved in the delivery/extension/fusion stage of autophagic degradation. This was further validated by a quantitative qRT-PCR of *ATG8A* and a few other autophagy core genes (Appendix A). Consistent with the RNA sequencing result, the expression of *ATG8A* was significantly reduced in the *lon1* mutant. Moreover, transcription factors *TGA1*, *TGA9*, and *TGA10* which were found to participate in ATG8 transcription were significantly downregulated (Figure 1A).

To investigate if the downregulation of autophagic core proteins affects autophagic flux, we crossed *lon1* mutant lines with a homogenous-single *copy-pATG8A:GFP-ATG8A* (*pATG8A* stands for *ATG8A* native promoter) stable transformation line in the *Col-0* background [22]. The *pATG8A:GFP-ATG8A* line had wild-type-like phenotypes and responded similarly to stress. Introducing *pATG8A:GFP-ATG8A* had no discernible effects on plant morphology or physiology. Two generations of hygromycin resistance screening were performed to introduce homogenous *pATG8A:GFP-ATG8A* into the *lon1* genetic background. *pATG8A:GFP-ATG8A* in both *Col-0* and *lon1* backgrounds was used for *ATG8A* expression and biochemical autophagic flux assay (Figure 1B,C). *ATG8A* expression and autophagic flux assays by Western blotting found much less GFP-ATG8A and almost invisible free GFP in *lon1* lines. Collectively, the results implicate a general ATG8A-associated autophagic flux reduction in the *lon1* mutant due to the downregulation of *ATG8A* and other autophagy core genes.

### 2.2. Lon1 Loss of Function Changes in Relative Proportion of Cellular Compartments at Level of Protein Abundance

The downregulation of autophagy flux in *lon1* mutants, as discovered in this study, along with the reported upregulation of unfolded protein responses [20], can lead to changes not only in the abundance of specific proteins but also in the relative proportion of cellular compartments in plants. To investigate the possibility of changes in the relative proportion of cellular compartments, we conducted label-free quantitative mass spectrometry using root total proteins (Appendix A). The relative proportion was determined by summing up the abundance of proteins in specific localizations defined by Subalive [23] over the total proteins. The results revealed that only the relative proportion of mitochondrial proteins was significantly higher in two *lon1* mutant lines compared to *Col-0* (Figure 2A). Conversely, the relative proportion of Golgi apparatus and vacuolar proteins decreased in both *lon1* mutant lines. ER proteins increased in *lon1-1* but decreased in *lon1-2*. There were no differences observed in the relative proportion of abundance for plastids and peroxisome proteins between *lon1* mutants and *Col-0*. The abundance of proteins in other organelles showed changes in only one allele but not in the other.

Regarding specific proteins, we examined mitochondrial proteases and reported mitophagy regulatory proteins [3,24]. Apart from the decrease in the abundance of mitochondrial processing peptidase-α2a (MPP-α2a) in *lon1* mutants, the abundance of other mitochondrial proteases significantly increased. Notably, FtsH10, intermediate cleavage peptidase (ICP55), and inner membrane protease (IMP2) exhibited a substantial increase (Figure 2B). Although autophagy flux reduction might downregulate mitochondria degradation, three receptors showed an increasing trend in *lon1* mutants (Figure 2B). Furthermore, we conducted a correlation analysis between changes in the levels of transcripts and their corresponding proteins or specifically for proteins localized to the mitochondria (Appendix A). The analysis revealed a correlation coefficient of 0.34 for total proteins and 0.37 for mitochondria-localized proteins, indicating a moderate positive correlation between the two variables. This suggests that transcriptional responses in *lon1* mutant lines contribute to a proportion of changes observed at the protein level.

### 2.3. Unfolded Protein Responses Are Not Associated with Autophagy Flux Reduction in lon1

Autophagic flux reduction in *lon1* mutant lines can contribute to unfolded protein responses. Only one isoform of ATG5 exists in the *Arabidopsis* genome; its knockout mutants are known to have deficient autophagy due to failed ATG8 lipidation [25]. Thus, we conducted RNA sequencing in an *atg5* mutant line *atg5-1* and compared the changes at the transcript level with *lon1-2*. Additionally, we constructed the *lon1-2atg5-1* double mutant in this study to investigate the combined loss of both Lon1 and autophagy. RNA deep sequencing analysis was compared among *lon1-2*, *atg5-1*, *lon1atg5*, and *Col-0* (Appendix A). Principal component analysis (PCA) suggests clear distinguishable changes at the transcript level for *lon1-2* and *atg5-1*, while the double mutant shows a combination of parental alleles (Figure 3A).

Furthermore, we extracted mitochondrial proteases and reported unfolded protein responses genes in *lon1* [20] to investigate how dysfunctional autophagy contributes to transcripts responses in *lon1*. The *atg5* mutant was utilized to investigate whether impaired autophagy elicits mitochondrial unfolded protein responses, illuminating the potential contribution of diminished autophagic flux in *lon1* mutants to unfolded protein responses. We found that the majority of mitochondrial proteases reported that UPR^mt^ [26,27], UPR^cp^ [28], and UPR^er^ [29] target genes are significantly upregulated in *lon1-2* and *lon1-2atg5-1* (Figure 3B–E, Appendix A), particularly pronounced in *lon1-2* which was reported recently [20]. However, this phenomenon was not observed in *atg5-1*, demonstrating that the absence of ATG5 does not significantly induce UPR^mt^, and the downregulation of autophagy levels in the *lon1* mutant is not the main cause of inducing UPRs or mitochondrial protease upregulation. Only a few genes, such as *ACS4* and *DTX3*, show the same responses in *lon1-2* and *atg5-1*, with over two-fold changes.

### 2.4. The Double Mutant, with Dysfunctional Lon1 and Autophagy, Exhibits an Additive Phenotype of Both Parental Lines

Although *lon1-2* and *atg5-1* exhibited distinct changes at the transcriptome level at the seedling stage, we found that *lon1-2* and *atg5-1* can cause similar phenotypes at the floral and seed developmental stages (Figure 4 and Figure 5).

While the growth retardation of the rosette at the seedling stage was observed in *lon1-2* and *atg5-1* [30,31], mutant lines reached the level of *Col-0* at five weeks old (Figure 4). It is notable that the height of *Arabidopsis* was lower in *lon1-2* and *atg5-1* compared with *Col-0* at the bolting stage. The number of senescent leaves was higher only in *atg5-1* but not in *lon1-2* compared with *Col-0*. The phenotype of the double mutant exhibits an additive effect from both parental lines: the plant height is notably lower than both *lon1-2* and *atg5-1*, while the number of rosette senescence leaves is close to that of *atg5-1* (Figure 4).

Developing seeds were observed and compared between the single mutant, the double mutant, and *Col-0* under microscopy after seed clearing treatment (Figure 5A). Both *lon1-2* and *atg5-1* showed retardation in embryo development; the embryo just reached heart or late heart development in the mutant lines when the *Col-0* embryo was already mature. The number of seeds in a silique was much lower, but the number of abnormal seeds with a twisted shape increased in *lon1-2* and *atg5-1* compared with *Col-0* (Figure 5B,C). For mature seeds, protein nutrients are stored in special vesicles called protein storage vacuoles (PSVs). The imaging of embryonic cotyledons within dry seeds by autofluorescence revealed that *Col-0* has regularly distributed PSVs, scattered quite evenly, while PSVs in *lon1-2* and *atg5-1* are clearly irregularly distributed. The distribution of PSVs in *lon1-2atg5-1* is even more irregular, with varying sizes mimicking both *lon1-2* and *atg5-1* (Figure 5D). Starch grains represent temporary storage during *Arabidopsis* embryo development and are eventually converted into oil bodies. Embryos with Nile red staining was performed to visualize oil bodies. The imaging showed that regular and very distinct oil bodies could be detected in *Col-0* embryos. However, *lon1-2* and *atg5-1* had significantly reduced and irregular oil bodies, and the phenotype of *lon1-2atg5-1* is a combination of both parental lines (Figure 5D).

### 2.5. Common Downregulation of BR Biosynthesis Genes in Single and Double Mutant Lines

We have observed several common phenotypes in *lon1-2*, *atg5-1*, and *lon1-2atg5-1* mutants compared to *Col-0*. These include delayed embryo development, a reduced number of seeds per silique, an increased proportion of abnormal seeds, and an uneven distribution of PSVs and oil bodies. To investigate if these phenotypes were contributed by the same transcriptome responses, we performed a GO functional and KEGG pathway enrichment analysis of differential expression genes (DEGs) in *lon1-2*, *atg5-1*, and *lon1-2atg5-1* (Figure 6 and Appendix A).

We found that 166 genes were significantly downregulated (FC < 1.5^−1^, *p* > 0.05) in all three mutant lines (Appendix A). It is noticeable that a significant enrichment of biosynthetic pathways and homeostasis was related to BRs and their precursors (squalene, triterpenoid, and sterol) (Figure 6A,B). BRs are widely present in the plant kingdom, and different types of BRs have been detected in both lower and higher plants [32], playing a crucial regulatory role, especially in embryo development, seed size, quality, and shape [33]. Analyzing the genes that are commonly downregulated in the three mutant lines collectively offers the advantage of clearly identifying the genes commonly downregulated across the mutants, thereby elucidating those involved in BR biosynthesis and homeostasis. Genes involved in BR biosynthesis include a series of Cytochrome P450 family members (*CYP87A2*, *CYP702A6*, *CYP716A1*, *MAB16.8*, *CYP716A2*, *CYP708A2*), as well as genes involved in BR precursor biosynthesis such as *BAS*, *PEN3*, *MRN1*, and *THAS1*, all of which are significantly downregulated in *lon1-2*, *atg5-1*, and *lon1-2atg5-1*. Similarly, genes related to BR homeostasis regulation, *BIA1/ABS1* and *BIA2*, show the same transcriptional levels, except for *BIA1* in *atg5-1* (Figure 6C). Overall, as a key plant hormone regulating seed development, genes involved in BR biosynthesis and homeostasis are significantly downregulated in all mutant types, which can contribute to the seed and embryo development deficiency phenotype in these three mutant types.

In addition to the obvious common downregulation of BR biosynthesis and homeostasis, we found that genes in the cell wall were commonly upregulated in both *lon1-2* and *atg5-1* (Appendix A). The common responses at the level of transcripts suggest dysfunctional Lon1 or autophagy shared specific feedback signaling. In contrast, we found that genes involved in endomembrane system organization, lytic vacuole, and cysteine-type endopeptidase activity were downregulated in *lon1-2* but upregulated in *atg5-1* (Appendix A). Genes in response to chitin were only upregulated in *lon1-2* but downregulated in *atg5-1* (Appendix A). The contrasting responses observed at the transcript level could indicate Lon1- or autophagy-specific reactions that are contradictory to each other.

## 3. Discussion

### 3.1. Autophagic Flux Downregulation Likely Contributes to Increases in Relative Proportion of Mitochondrial Proteins to Maintain Steady Respiration in lon1 Mutant Lines

Transcriptome data showed that most autophagy core genes were downregulated in *lon1-2* (Figure 1A). The positive regulators of autophagy transcription factors *TGA1*, *TGA9*, and *TGA10*, which transcribe *ATG8B* and *ATG8E* [34], were significantly downregulated. This was further evidenced by a lower level of both GFP-ATG8a and free GFP in the *lon1* backgrounds (Figure 1B). This suggests core autophagy factors were downregulated at both the transcription and the protein level to downregulate autophagic flux in the *lon1* background. The relative proportion of vacuolar proteins displays a downregulation resembling that seen in autophagy mutants (Figure 2A), as reported by Li et al. [21]. This finding further strengthens the evidence supporting the downregulation of autophagic flux in *lon1* mutant lines. Autophagy degradation contributes to basal-level mitochondrial protein degradation and organelle degradation in response to stresses or during developmental stage transition [15,19,21]. Autophagy was found to be stimulated post-transcriptionally under mild and persistent UPR^er^ for either survival or programmed cell death [35,36] and also activated in the mitochondrial protease FtsH4 loss-of-function mutant which led to ROS overproduction and the early senescence phenotype [37]. Autophagy downregulation, however, was not reported hitherto. No obvious changes in gross respiration were observed in the photosynthetic and non-photosynthetic tissue of *lon1* mutant lines [38], in spite of its large-scale change in mitochondrial proteome and morphology. However, a pronounced upregulation of mitochondrial proteins was revealed in *lon1* lines (Figure 2). This was found to be contributed at least partially by the UPR^mt^ responses reported by us recently [20], evident by a moderate positive correlation between transcripts and their encoded protein abundance (Appendix A). In this study, we discovered a general downregulation of autophagic flux (Figure 1 and Figure 2), which can lower the degradation of mitochondria and potentially contribute to mitochondria accumulation observed in *lon1* mutant lines. *lon1* plants may derive benefits from increased levels of mitochondrial proteins, ensuring stable respiration rates and providing essential energy for growth and development. This hypothesis gains support from the observation that, despite a notable decrease in electron transport chain complexes, *lon1* mutant lines did not exhibit significant changes in gross respiration [38]. Meanwhile, the general reduction in autophagic flux cannot present the downregulation of the selective degradation of mitochondria by mitophagy. A targeted selective mitochondrial autophagic degradation assay in *lon1* mutant lines would help resolve this uncertainty in a future study. Moreover, we observed a clear upregulation of mitochondrial proteases at both the transcriptional and protein levels (Figure 2B and Figure 3B). This supports a distinct functional compensation effect from other mitochondrial proteases, in addition to the unfolded protein response reported recently by us [20]. Taken together, the compensatory effect from the upregulation of UPR target proteins and mitochondrial proteases, coupled with the downregulation of autophagic flux, can collectively contribute to maintaining mitochondrial protein homeostasis in the context of an increase in the relative proportion of mitochondria in the absence of functional Lon1.

### 3.2. Unveiling the Role of Lon1 and Autophagy in BR Biosynthesis and Seed Development

Autophagy plays a crucial role in nutrient recycling, particularly in seed development in plants, and contributes to nitrogen reactivation at the whole-plant level. Compared to wild-type (WT) plants, autophagy mutants display abnormal carbon and nitrogen content in seeds. The *Arabidopsis atg5* mutant displays several typical phenotypes observed in other autophagy mutants, such as smaller rosette leaf size, heightened sensitivity to nitrogen and carbon starvation, decreased yield, and developmental abnormalities in seeds [39]. The disparity in seed quality between autophagy mutants and wild-type (WT) plants may stem from the significant impact of autophagy mutations on carbon and nitrogen metabolism, including processes like photosynthesis and nitrogen reactivation [40]. In comparison to WT, *atg5* seeds demonstrate premature browning and a higher incidence of seed abortion. These observations suggest an accelerated seed development in *atg5*, accompanied by an aberrant or incomplete pathway for protein storage deposition [41]. In this study, we observed poor seed and embryo development, as well as an uneven distribution of PSVs and oil bodies, which were common phenomena in the *lon1-2*, *atg5-1*, and *lon1-2atg5-1* mutants (Figure 4 and Figure 5). Moreover, these phenotypes were exacerbated in the double mutants, indicating an additive effect of both parental lines.

Subsequently, we analyzed the transcriptome results to identify commonalities among them. Through a GO and KEGG enrichment analysis of genes significantly downregulated in all three mutants, the results consistently pointed to BR as the plant hormone involved. The biosynthesis and signal transduction of BRs have been shown to regulate various aspects of seed development, including seed size, number, germination, and composition in rice [42]. BRs play a crucial role in determining the size, quality, and shape of *Arabidopsis* seeds. Specifically, BRs regulate seed size and shape by modulating specific seed development pathways at the transcriptional level. Seeds of the BR-deficient mutant de-etiolated2 (*det2*) are characterized by being smaller, less elongated, exhibiting delayed embryonic development, and having reduced embryo cell size and number compared to seeds of wild-type plants [33]. The specific genes enriched included several genes from the P450 family involved in biosynthesis, as well as genes such as *BAS*, *PEN3*, *MRN1*, and *THAS1* linked with BR precursor synthesis. Additionally, genes engaged in homeostasis regulation such as *BIA1* and *BIA2* were significantly downregulated in all three mutants (Figure 6). The catalytic enzymes involved in crucial steps of BR biosynthesis have been mostly elucidated, with the majority belonging to the cytochrome P450 family [43]. A shared downregulation of the BR biogenesis pathway implies the potential existence of a common mediator for situations where mitochondrial Lon1 degradation or autophagy degradation is impaired. Although a direct mechanistic link between the dysfunction of two degradation pathways and BR biosynthesis remains elusive, further investigations utilizing BR biosynthesis mutants or overexpression mutant lines, in conjunction with BR treatment in associated mutant lines, may shed light on the missing connection between the downregulation of BR biosynthesis and the phenotype formation in *lon1*, *atg5*, and their double mutant.

## 4. Materials and Methods

### 4.1. Preparation of Arabidopsis Plants

*Arabidopsis* (Columbia-0 ecotype) was cultivated in soil under a 16/8 h light/dark cycle at room temperature (18–22 °C), with a light intensity of 120–150 µmol/m^2^. *Arabidopsis* mutant lines, including the *lon1-1* point mutation (AT5G26860, site 809 termination) and the T-DNA insertion mutant *lon1-2* (AT5G26860, SALK_012797), T-DNA insertion mutants *atg5-1* (AT5G17290, SAIL_129_B07), and the double mutant lines were grown under the same conditions as *Col-0*. The *Arabidopsis* stable transformation line *pATG8A:GFP-ATG8A* which used the ATG8A native promoter [22] was used for autophagic flux assay in both *Col-0* and *lon1* mutant lines. Two generations of hygromycin resistance screening were performed to introduce homogenous *pATG8A:GFP-ATG8A* into the *lon1* genetic background.

### 4.2. Protein Sample Extraction

Snap-freeze 100–250 mg of tissue in liquid nitrogen, and grind using a mortar and pestle (2–8 mg protein from 250 mg fresh weight of tissue). Transfer the ground tissue to 2 mL Eppendorf tubes sitting in liquid nitrogen. Add 400 μL of extraction buffer, and vortex vigorously. The mixture should have a slurry consistency; add a little more extraction buffer, vortex, place tubes on their side, and rock on ice for 10 min. Centrifuge at 10,000× *g* for 5 min at 4 °C, then move the supernatant to new tubes, and discard the pellet. To the supernatant, add 800 μL of methanol, vortex, then add 200 μL of chloroform, vortex, and finally add 500 μL of ddH_2_O, vortex. Centrifuge for 5 min at 10,000× *g* at 4 °C. Discard the upper aqueous phase carefully so as to not remove the protein layer at the interface. Add 500 μL of methanol, vortex, and centrifuge for 10 min at 9000× *g* at 4 °C. Discard the supernatant. Add 1 mL of pre-cooled (−20 °C) 90% [*v*/*v*] acetone to the resulting pellet, vortex vigorously, and incubate for 1 h at −20 °C. Centrifuge for ten minutes at 14,000× *g* at 4 °C. Repeat this step by adding another 1 mL of pre-cooled (−20 °C) 90% [*v*/*v*] acetone to the resulting pellet, vortexing vigorously and incubating for 1 h at −20 °C. Centrifuge again for ten minutes at 14,000× *g* at 4 °C. Discard the supernatant. Dry the pellet in a fume hood; reverse the tube for about 5–10 min until no visible liquid remains in the tube. Resuspend the pellet, or alternatively, store the pellet in a −80 °C freezer.

### 4.3. Label-Free Quantification by Mass Spectrometry

A total of 30 μg of root protein isolated from *Col-0* and *lon1* mutant lines was dissolved in a resuspension buffer containing 7 M Urea, 2 M Thiourea, 50 mM NH_4_CO_3_, and 10 mM DTT. The proteins were subjected to in-solution digestion using trypsin. Tryptic peptides resulting from these in-solution digestions were dehydrated using a centrifugal vacuum concentrator (Labconco, Kansas city, MO, USA). C18 spin columns Empore321069D (Sigma-Aldrich, St. Louis, MO, USA) were employed for desalting prior to LC-MS/MS analysis. An LC-MS/MS analysis was conducted using a Dionex Ultimate 3000 nano-HPLC system coupled with an Orbitrap Exploris 240 mass spectrometer (Thermo Fisher Scientific, Wilmington, DE, USA). The dry samples were processed in a speed vacuum and then resuspended in 0.1% formic acid (FA) to achieve a concentration of 1 µg/μL, making them ready for mass spectrometry analysis. EASY-Spray columns (50 μm i.d. × 15 cm and 75 μm i.d. × 50 cm) were packed in-house with 2 μm Acclaim PepMap RSLC C18 beads (Thermo Fisher Scientific, Wilmington, DE, USA). The LC-MS/MS analysis followed a gradient profile delivered at 300 nL/min: starting at 97% solvent A (0.1% formic acid in water) and transitioning to 10% solvent B (0.08% formic acid in 80% acetonitrile) over 5 min, followed by a transition from 10% to 50% solvent B over 2 h.

The raw data processing and database searches were conducted using Proteome Discoverer software 2.4 from Thermo Fisher Scientific. To match peptide fragments generated by mass spectrometry, the *Arabidopsis* database TAIR10 from the *Arabidopsis* Information Resource (TAIR) was utilized. PD Search parameters included a strict false discovery rate (FDR) of less than 0.01 and a relaxed FDR of less than 0.05. Static modifications involved carbamidomethyl (Cys), while dynamic modifications included methionine oxidation, protein N-terminal acetylation (for Arg, His, Ser, Thr, Tyr), and phosphorylation (for Asp, Glu, His, Lys, Arg, Ser, Thr), with a maximum allowance of 2 missed cleavage sites. Error tolerances for mass spectrometry (MS) were set at 10 ppm, and for MS/MS, they were set at 0.02 Da.

### 4.4. Total RNA Deep Sequencing

Total RNA was extracted from ten-day-old *Arabidopsis* seedlings grown on plates under long-day conditions using the Ambion mirVana miRNA Isolation Kit (Thermo Fisher Scientific, Wilmington, DE, USA), following the manufacturer’s protocol. The quality of the RNA was assessed using the Agilent 2100 Bioanalyzer from Agilent Technologies, Santa Clara, CA, USA. Transcriptome sequencing and analysis were performed by OE Biotech Co., Ltd. in Shanghai, China. Samples with an RNA Integrity Number (RIN) of 7 or higher were selected for subsequent analysis. Library construction was carried out using the TruSeq Stranded mRNA LT Sample Prep Kit from Illumina, San Diego, CA, USA, following the manufacturer’s instructions. These libraries were then sequenced on the DNBSEQ-T7 Illumina sequencing platform (Illumina, San Diego, CA, USA), generating 125 bp/150 bp paired-end reads.

The raw data (raw reads) underwent processing using Trimmomatic (version 0.36). This involved the removal of reads containing poly-N sequences and low-quality reads to obtain clean reads. Subsequently, the clean reads were aligned to the reference genome using hisat2 (version 2.2.1.0). The Fragments Per Kilobase of transcript per Million mapped reads (FPKM) value for each gene was calculated using cufflinks [44], and the read counts for each gene were obtained using htseq-count [45]. Differentially Expressed Genes (DEGs) were identified with the DESeq R package (version 2.8), employing functions such as “estimate Size Factors” and “nbinomTest”. A significance threshold was set with a *p*-value less than 0.05 and a fold change greater than 1.5. To explore gene expression patterns, a hierarchical cluster analysis of DEGs was performed. Gene Ontology (GO) enrichment and Kyoto Encyclopedia of Genes and Genomes (KEGG) pathway enrichment analyses of DEGs were conducted using R, based on the hypergeometric distribution.

### 4.5. RNA Extraction and Q-PCR Analysis

Ten-day-old seedlings from *Col-0* and two *lon1* mutant lines, grown under long-day conditions, were collected for RNA extraction. The collected seedlings, approximately 0.1 g in weight, were rapidly frozen in liquid nitrogen and then ground into a fine powder using 2 mm beads and a homogenizer. RNA was extracted using the TaKaRa (9769) MiniBEST Plant RNA Extraction Kit (Shiga, Japan), following the manufacturer’s instructions. For cDNA synthesis, 500 ng of RNA was utilized, employing the TSINGKE (TSK302M) Goldenstar™ RT6 cDNA Synthesis Kit Ver.2 (Beijing, China). Transcripts of selected genes were quantified using TaKaRa (RR420A) TB Green^®^ Premix Ex Taq™ (Tli RNaseH Plus) (Shiga, Japan) and the Realplex2 system from Eppendorf (Hamburg, Germany). The Q-PCR data were normalized to housekeeping genes, specifically *UBQ10*, before making comparisons across different ecotypes. All primers used for Q-PCR analyses are listed in Appendix A.

### 4.6. Autophagic Degradation Assay by Western Blotting

Ten-day-old plate-grown *Arabidopsis* seedlings were harvested and homogenized in liquid nitrogen to a fine powder. Total proteins were extracted with a protein extraction buffer (1 M Tris pH 8.0, 5% SDS, 20% glycerol, 10% β-mercaptoethanol, 1% protease inhibitors cocktail). Protein samples in the extraction buffer were centrifuged at 12,000× *g* for 30 min to precipitate insoluble material. The concentration of total protein in the supernatant was determined using Amidoblack (Biotopped, Beijing, China). A total of 30 µg of proteins was separated by SDS-PAGE. Proteins were transferred to a PVDF membrane and incubated in a mouse polyclonal antibody against GFP (antibody, UM3002). A goat anti-mouse antibody (HRP) ZSGB-BIO, ZB-2305 (Beijing, China) was used for visualization.

### 4.7. Visualization of Embryo, PSV, and Oil Body Using Microscopy

To visualize embryos, begin by harvesting siliques from *Arabidopsis thaliana* plants after the flowers have wilted. Carefully open the siliques using forceps and a needle attached to a syringe. Extract the seeds and immerse them in a clearing agent. Allow the seeds to soak at room temperature overnight, adjusting the duration based on their developmental stage and seed color (typically 12 h for white seeds and over 24 h for green seeds). Once the seeds become fully transparent, photograph them under a high-resolution upright microscope, preferably using DIC mode for the optimal visualization of colorless transparent objects and achieving a realistic three-dimensional relief effect.

To visualize protein storage vacuoles (PSVs), utilize autofluorescence in dissected dry seeds following water absorption. Acquire autofluorescence images of the seeds using a Leica TCS SP5 scanning confocal microscope (Wetzlar, Germany), employing 488-nanometer light and filters ranging from 500 to 550 nanometers. Process the acquired images using Leica DM6000 CFS software (https://w2.uib.no/filearchive/leica-dm6000-cfs.pdf, accessed on 11 May 2024), converting them into TIFF files for further analysis and presentation. For observing oil bodies, extract embryos onto a coverslip after the dry seeds have absorbed water. Stain the embryos with Nile red stain solution prepared at a concentration of 1 μg/mL in anhydrous methanol, ensuring staining occurs in darkness for at least 20 min. Rinse the stained embryos with distilled water multiple times and proceed to observe the oil bodies. Utilize a Leica TCS SP5 scanning confocal microscope, employing 561-nanometer light and filters ranging from 560 to 630 nanometers for optimal image capture. Process the obtained images using Leica DM6000 CFS software, converting them into TIFF files for analysis and presentation purposes.

## 5. Conclusions

This study delved into the regulation of autophagy in the *Arabidopsis* Lon1 knockout mutants. Our findings unveiled a notable downregulation of core autophagy genes, leading to a decrease in autophagic flux in the Lon1 mutants. This downregulation of autophagic flux potentially contributes to an increase in the relative proportion of mitochondrial proteins, which is crucial for maintaining steady respiration in plant cells. Furthermore, we discovered that the mitochondrial unfolded protein response in the Lon1 mutant is not driven by decreased autophagic flux. The generation of a double mutant (*lon1-2atg5-1*) exhibited a combined phenotype of the single mutants, supporting the independent yet potentially interconnected roles of Lon1 and autophagy in plant development. Both single mutants and the double mutant displayed abnormal seed development and reduced seed number. Transcriptome analysis identified a significant suppression of genes associated with brassinosteroid (BR) biosynthesis and homeostasis in all three mutant lines, potentially contributing to the observed seed and embryo developmental defects. These findings collectively suggest that while the mitochondrial protease Lon1 and autophagy function independently in phenotype formation and transcriptome responses, the two degradation pathways can interact to maintain mitochondrial protein homeostasis.

## Figures and Tables

**Figure 1 ijms-25-05425-f001:**
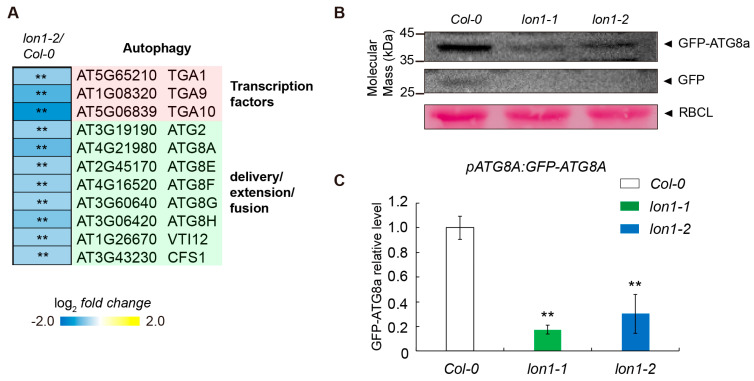
The disruption of *Arabidopsis* Lon1 downregulates autophagic flux by reducing the expression of autophagy core genes. (**A**) Log-transformed fold changes in the transcript levels of genes encoding transcription factors (*TGA1*, *TGA9*, and *TGA10*), autophagy core proteins involved in autophagy membrane delivery (*ATG2*), phagophore extension (*ATG8A*, *E*, *F*, and *G*), and tonoplast fusion (*VTI12* and *CFS1*) are presented. All displayed genes exhibit a significant decrease (FC < 1.5^−1^, *p* < 0.05, Student’s *t*-test) in *lon1-2* compared with *Col-0*. (**B**,**C**) Autophagic flux assay in 7-day-old seedlings measured by Western blotting, represented by the difference in GFP-ATG8 between *Col-0* and *lon1* mutants. The GFP-ATG8 relative protein level in *lon1* mutant lines was acquired by normalizing to *Col-0*. Ponceau staining serves as a control for equal protein loading. Error bars represent standard deviations from three biological replicates. Statistical significance was determined using Student’s *t*-test (** indicates *p* < 0.01).

**Figure 2 ijms-25-05425-f002:**
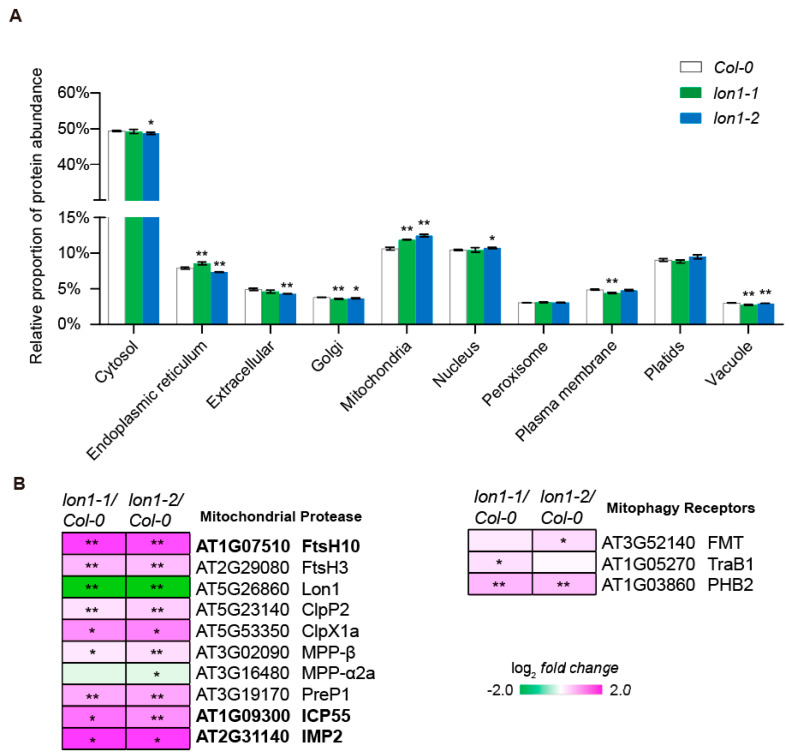
Lon1 loss of function modulates relative proportion of proteins in cellular compartments. (**A**) Relative proportion of proteins in different subcellular compartments obtained by label-free quantification (LFQ) mass spectrometry analysis using root total proteins extracted from ten-day-old *Col-0* and *lon1* lines. Error bars represent standard deviations from three biological replicates. Statistical significance was determined using Student’s *t*-test (* indicates *p* < 0.05, ** indicates *p* < 0.01). (**B**) Log-transformed fold changes in protein abundance of mitochondrial proteases and mitophagy-associated proteins in *lon1* mutants compared with *Col-0* (* indicates *p* < 0.05, ** indicates *p* < 0.01). Proteins with fold changes greater than 2 in *lon1* lines are highlighted in bold fonts.

**Figure 3 ijms-25-05425-f003:**
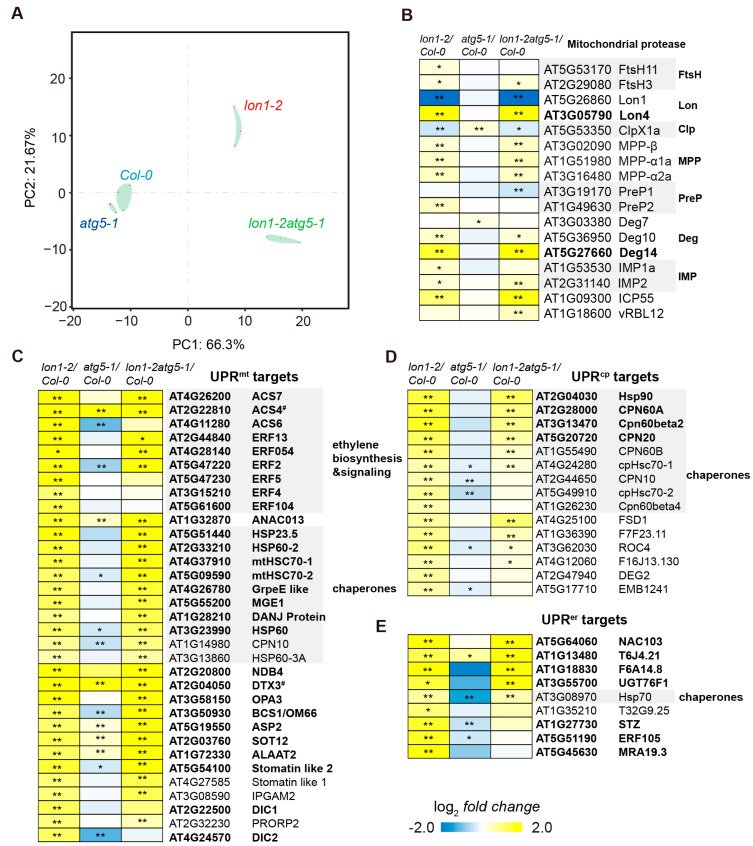
RNA deep sequencing reveals independent changes in transcript levels between *lon1-2* and *atg5* mutant lines. (**A**) Principal component analysis (PCA) performed using RNA deep sequencing data from *Col-0*, *atg5-1*, *lon1-2*, and *lon1-2atg5-1*. (**B**) Log-transformed fold changes in transcript levels of mitochondrial proteases in mutant lines compared with *Col-0* are presented. (**C**–**E**) Log-transformed fold changes in transcript levels showing statistically significant induction (FC > 1.5, *p* < 0.05) in *lon1-2* for mitochondrial unfolded protein responses, including UPR^mt^ [26,27], UPR^cp^ [28], and UPR^er^ [29] target genes that are extracted and presented. Transcripts with fold changes greater than 2 in *lon1-2* are highlighted in bold. The symbol # denotes transcripts with fold changes greater than 2 in *lon1-2*, *atg5-1*, and *lon1-2atg5-1*. Genes encoding mitochondrial proteases, ethylene biosynthesis and signaling proteins, and chaperones are annotated. Blue-yellow color gradient represents the log_2_*FC* values between mutants and *Col-0*. * indicate *p* < 0.05, ** indicate *p* < 0.01.

**Figure 4 ijms-25-05425-f004:**
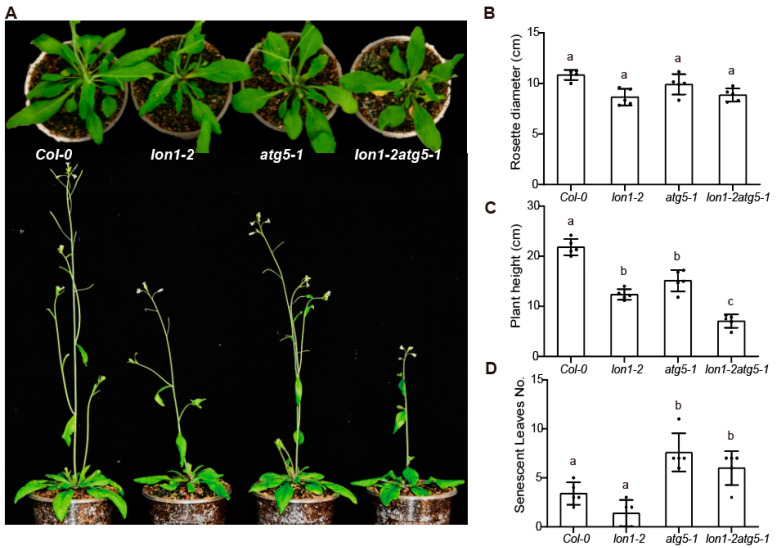
*lon1-2atg5-1* double mutant exhibits additive growth phenotype compared to single mutants. (**A**) Presentation of five-week-old soil-grown *Arabidopsis* plants of *Col-0*, *lon1-2*, *atg5*, and *lon1-2atg5*. (**B**–**D**) Measurement and presentation of rosette diameter, plant height, and numbers of senescent leaves as column graphs. Error bars indicate standard deviation across five biological replicates. Different letters in grouping of different lines indicate statistically significant differences based on one-way ANOVA test. The dots represent biological replicates.

**Figure 5 ijms-25-05425-f005:**
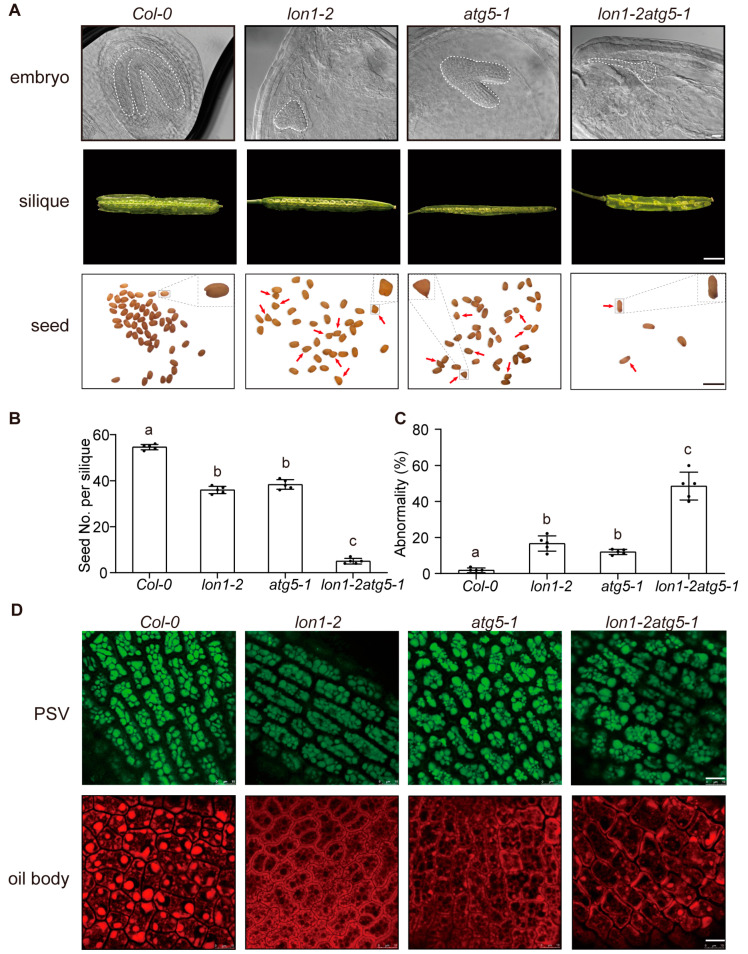
*lon1-2atg5-1* double mutant displays increased severity in abnormal seed development and nutrient storage. (**A**) Observations reveal embryo development retardation, reduction in number of seeds per silique, and seed shape changes in *lon1-2*, *atg5*, and *lon1-2atg5* compared with *Col-0*. Red arrows indicate abnormal seeds. Scale bar indicates 20 μm for embryos, 2 mm for siliques, and 1 mm for seeds. (**B**,**C**) Number of seeds per silique and abnormality in seed shape are displayed for *Col-0* and mutant lines. Different letters in grouping of different lines indicate statistically significant differences determined by one-way ANOVA. The dots represent biological replicates. (**D**) Observation of protein storage vacuoles (PSVs) with autofluorescence and Nile red-stained oil bodies in cotyledons using fluorescence confocal microscopy. Scale bar indicates 10 μm.

**Figure 6 ijms-25-05425-f006:**
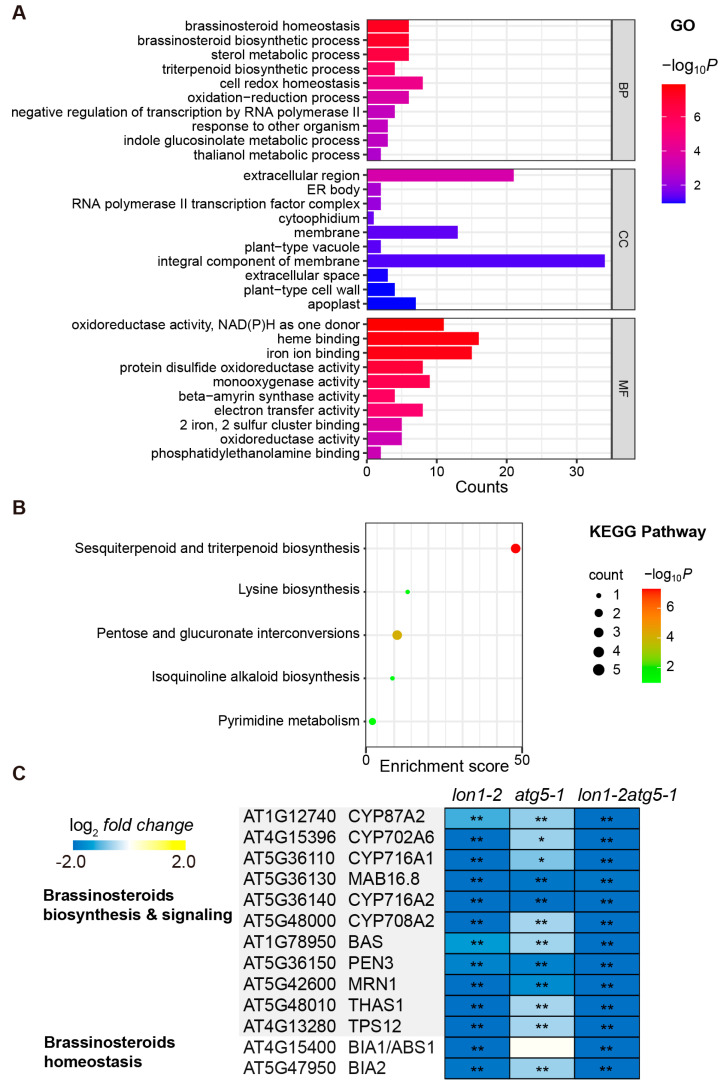
Commonly downregulated genes in *lon1-2*, *atg5-1*, and *lon1-2atg5-1* are enriched with the brassinosteroid biosynthesis pathway. (**A**) A total of 166 genes were commonly downregulated in *lon1-2*, *atg5-1*, and *lon1-2atg5-1* and were subjected to an enrichment of Gene Ontology (GO) analysis. The horizontal bars represent the counts for biological processes (BPs), cell compartments (CCs), and molecular functions (MFs), while the blue–red color bar indicates the range of Log-transformed *p* values. (**B**) The KEGG pathway enrichment analysis of downregulated genes in all three mutants. The size of the bubbles corresponds to the number of genes involved, while the green–red color bar indicates the range of Log-transformed *p* values. (**C**) Log-transformed fold changes in transcript levels for genes related to brassinosteroids in *lon1-2*, *atg5-1*, and *lon1-2atg5-1* (* indicates *p* < 0.05, and ** indicates *p* < 0.01).

## Data Availability

Project Name: *Arabidopsis* Lon1 disruption decreases autophagy flux affecting seed development through brassinosteroid regulation. Project accession: PXD050885. RNA-seq data GSE262650 accessed on 11 May 2024.

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
