# Peer review of "Lon1 Inactivation Downregulates Autophagic Flux and Brassinosteroid Biogenesis, Modulating Mitochondrial Proportion and Seed Development in Arabidopsis"

_ijms, 2024, doi:10.3390/ijms25105425_

Round 1
Reviewer 1 Report
Comments and Suggestions for Authors
The manuscript of Ce Song and co-authors is devoted to the study of crosstalk between these two degradation pathways - mitochondrial proteases and cytosolic autophagy. Using three types of mutant plants, the authors registered a number of effects. However, the manuscript lacks a general conclusion and it is difficult to understand whether the data obtained indicate a connection between these degradation systems. And if the connection exists, what is its biological meaning? In my opinion, the authors should reconsider their results and resubmit the manuscript after a thorough revision.
The main comments
1) The title of the manuscript does not reflect all the results obtained.
2) The abstract lacks a clear purpose of the study and a general conclusion on the results obtained.
3) Materials and methods section.
- 4.1 There is no information about all mutants. How the plants with pATG8A:GFP-ATG8A were obtained.
- 4.3 How many biological and technical replicates were there?
-4.6 Why were control and mutant plants with pATG8A:GFP-ATG8A and anti-GFP antibodies used to assess autophagy, rather than just mutant plants and anti-ATG8 antibodies? How did the introduction of pATG8A:GFP-ATG8A affect the properties of plants? How were the results normalized?
4) Results and Discussion sections.
- If the authors found downregulation of ATG8 in lon mutant (Fig S1), why was the ATG5 mutant, but not ATG8 mutant used?
- From all the data presented, it is unclear what the cooperation of degradation pathways is and what is the meaning of this cooperation.
- Why does the absence of Lon1 lead to downregulation of autophagy? What is the biological meaning.
- Regarding the Lon1 mutant. There are other Lon-proteases in plants, for example, from the authors’ results it follows that upregulation of Lon4 takes place in lon1-2 mutant (Fig. 3). What about compensation effects?
- Figure 1 (original). What is the reason for the nonspecific staining? There are three stripes next to each other, how did the authors understand that the stripe in the middle is GFP-ATG8? Can these results be trusted? Why do the authors think it is possible to evaluate autophagy not only by GFP-ATG8, but also by GFP?
- Figure 1. The title is incorrect, since data is presented not only about transcripts. “Total proteins were immunoblotted with anti-GFP antibodies” is not correct phrase. There is no normalization of Western blotting results.
- Data S1. %coverage, unique peptides and other - these are averaged results or results from one sample? Were there repetitions?
- About the double mutant and the additive effect (Fig 4). In my opinion, if you turn off any two key genes in a plant, the effect in the double mutant will be higher than in the single mutant. But this is not proof of cooperation between systems.
- Figure 3. What is shown in green and pink in all cases? This is not clear everywhere. Maybe it’s better to bring proteins of the same class next to each other? What do other proteins correspond to (not chaperones and not proteins involved in ethylene biosynthesis)?
- Fig. 5. The abnormality of the seeds cannot be understood from the figure.
- Fig.6A,B. Why in the article, not in the supplement, are the results shown not for the three mutants separately, but in total? What is the significance of such a presentation of results?
- 3.2 The name is incorrect. How a proteolytic enzyme can regulate at the transcript level?
6) The manuscript does not contain a general conclusion from the results obtained, from which one could draw a conclusion about their significance.
Reviewer 2 Report
Comments and Suggestions for Authors
This paper is well written and reports well designed experiments resulting in important findings.
A few points need attention
Line 32. ‘highly regulated’: does this apply more to mitochondria than t other cell compartments?
Lines 142-143. Since the work here was on roots, ‘chloroplasts’ should be ’proplastids’, ‘etioplasts’ or (as in Figure 2) ’plastids’.
Line 309. Define ‘gross respiration’.
Lines 343-245. Presumably mention of ‘grains’ refers to grasses rather than Arabidopsis.
Reviewer 3 Report
Comments and Suggestions for Authors
The manuscript entitled "Disruption of Arabidopsis Lon1 Downregulates Autophagic Flux and Brassinosteroid Biogenesis" delves into the relationship between mitochondrial protein homeostasis and autophagy in plants, focusing specifically on the Lon1 protease pathway in Arabidopsis. It examines the effects of Lon1 mutations on autophagy-related processes and the synthesis of brassinosteroids, which are vital for plant development and stress response. The research is comprehensive, incorporating genetic mutations, transcriptomic, and proteomic analyses to explore the dynamics between Lon1 protease, autophagic flux, and brassinosteroid biogenesis. Employing both single and double mutants offers deeper insights into the genetic interactions and their impact on plant phenotypes and internal biochemical pathways. The objectives of this study are well-defined and the results are of considerable interest. I have no major concerns; however, I offer some suggestions below:
1. The study describes interactions between various proteins and genetic pathways affected by the Lon1 mutation. However, it could benefit from a more detailed explanation of how these interactions lead to the observed phenotypic changes, such as reduced autophagic flux and altered brassinosteroid synthesis. Expanding the mechanistic description to link these molecular events to physiological outcomes would enhance the clarity and impact of the findings. Further discussion or outlining future research plans could address this issue effectively.
2. Although the study presents extensive data on changes in gene expression and protein levels, there appears to be a gap in fully explaining the mechanistic link between these molecular changes and the observed physiological outcomes (e.g., abnormal seed and embryo development). Conducting additional experiments, such as functional assays or in vivo observations, would help to reinforce these connections.
3. While the manuscript demonstrates correlations between genetic mutations and changes in plant phenotypes, it lacks direct evidence to show that these genetic changes cause the phenotypes through the proposed mechanisms. Implementing functional assays, such as rescue experiments or specific inhibitor studies, would strengthen the argument that the observed molecular changes are directly contributing to the plant’s developmental abnormalities. Further discussion or future research plans would be beneficial in addressing this concern.
4. The complexity and volume of data, particularly from high-throughput sequencing and proteomics, might challenge readers in understanding the direct implications of the findings on plant biology. Although the study is methodologically sound, enhancing the connection between data presentation and biological interpretation by simplifying the presentation or providing more guided explanations could improve comprehension.
5. The manuscript is written with a high level of technical accuracy and specificity, suitable for its target audience of molecular biologists and plant scientists. The scientific terminology is used precisely and appropriately throughout the text. However, some sentences are overly complex, containing multiple clauses and extensive technical details. Simplifying these sentences into more digestible parts could enhance understanding and readability.
Comments on the Quality of English LanguageThe manuscript is written with a high level of technical accuracy and specificity, suitable for its target audience of molecular biologists and plant scientists. The scientific terminology is used precisely and appropriately throughout the text. However, some sentences are overly complex, containing multiple clauses and extensive technical details. Simplifying these sentences into more digestible parts could enhance understanding and readability.
Round 2
Reviewer 1 Report
Comments and Suggestions for Authors
I thank the authors for their detailed answers. The authors revised the manuscript and improved the perception of the results obtained and their significance. However, questions still remain regarding the data presented. The authors left in the title of the article only the results concerning the functional significance of Lon protease. At the same time, in the abstract, as background, the authors write about the possible cooperation of two degradation systems. This gets the impression that this is exactly what will be explored. The authors obtained some data about functional significance of Lon protease and some data indicating a possible connection between the two degradation systems, but requiring confirmation through additional experiments. Therefore, in my opinion, the authors should revise the article again, especially the title, abstract and conclusion, so that the general concept of the study is clearly understood.
Title.
The expression “Disruption of Arabidopsis Lon1” should be rephrased.
The abstract still requires revision. It is necessary to clearly indicate which plants (Arabidopsis) and mutants were used, what are lon1 and atg5, make the content consistent and write a general conclusion. The expression “These findings suggest an additive effect between lon1-2 and atg5-1” should be rephrased. This statement “These findings suggest an additive effect between lon1-2 and atg5-1, indicating separate functions of Lon1 and autophagy” is controversial.
Keywords do not contain anything about Lon proteases.
Authors should check the text for repetitions in the sentences, for example in the legend to Figure 1 and in lines 92-94.
Results and discussion
Below are comments on some of the authors' answers.
- Why were control and mutant plants with pATG8A:GFP-ATG8A and anti-GFP antibodies used to assess autophagy, rather than just mutant plants and anti-ATG8 antibodies? How did the introduction of pATG8A:GFP-ATG8A affect the properties of plants? How were the results normalized?
Response: Thank you for your inquiries. Firstly, the objective of the autophagic flux assay experiment was to evaluate whether the transcriptional downregulation of ATG8 isoforms, such as ATG8A, correlates with a corresponding decrease in ATG8A protein levels in lon1 compared to Col-0. Utilizing the pATG8A:GFP-ATG8A line, which harbors the ATG8A native promoter, was ideal for this purpose. Given that plants possess nine ATG8 isoforms (ATG8A-I), the ATG8 antibody lacks the specificity to distinguish between them. Therefore, we opted to employ the readily available pATG8A:GFP-ATG8A line in both Col-0 and lon1 genetic backgrounds, utilizing a more accessible GFP antibody for the autophagic flux assay.
Moreover, the stable transformation line of pATG8A:GFP-ATG8A exhibited phenotypes indistinguishable from the wild type (WT) under normal growth conditions and demonstrated similar responses to starvation and stress-induced conditions as the WT. Consequently, the
introduction of pATG8A:GFP-ATG8A exerted no discernible effects on plant morphology or physiology (see reference as following).
For western blotting, normalization was performed relative to ponceau staining bands of RBCL. A column graph depicting Student’s T-test results, along with error bars derived from three biological replicates, has been provided for clarity.
Reference
https://pubmed.ncbi.nlm.nih.gov/28878796/
This explanation should be briefly added to the text of the article.
- If the authors found downregulation of ATG8 in lon mutant (Fig S1), why was the ATG5 mutant, but not ATG8 mutant used?
Response: Thank you for your question. We employed the atg5 mutant to explore whether impaired autophagy triggers mitochondrial unfolded protein responses, shedding light on the potential contribution of reduced autophagic flux in lon1 mutants to unfolded protein responses. The choice of the atg5 mutant was dictated by the unavailability of an atg8 knockout mutant. Arabidopsis harbors nine isoforms of the ATG8 protein (ATG8a–ATG8i) with functional redundancy. Despite extensive investigations, researchers have yet to isolate homozygous mutants for all nine ATG8 isoforms. Conversely, ATG5 plays a pivotal role in ATG8 lipidation, and mutations in ATG5 result in diminished autophagosome formation.
Reference
https://pubmed.ncbi.nlm.nih.gov/20136727/
https://pubmed.ncbi.nlm.nih.gov/24947672/
This explanation should be briefly added to the text of the article. And what about ATG5 isoforms?
- Regarding the Lon1 mutant. There are other Lon-proteases in plants, for example, from the authors’ results it follows that upregulation of Lon4 takes place in lon1-2 mutant (Fig. 3). What about compensation effects?
Response: Thank you for your question. We wholeheartedly acknowledge the presence of compensation effects in the lon1 mutant. Our observations indeed reveal pronounced unfolded protein responses in lon1 mutants, characterized by the upregulation of numerous proteases and chaperones (Figure 3). Notably, these unfolded protein responses are intricately linked to mitochondrial retrograde signaling, particularly involving the transcription factor ANAC-017, as recently elucidated by our team (Reference: "Protein aggregation in plant mitochondria inhibits translation and induces an NAC017-dependent ethylene-associated unfolded protein response. bioRxiv 2023, 10.1101/2023.01.11.523570"). Although our transcriptome data identified Lon4 upregulation in lon1-2, Lon4 was notably absent in the proteomics data. Lon4 has long been recognized as a gene that is expressed but fails to undergo correct translation in plants. Hence, we posit that Lon4 may primarily respond at the transcriptional level.
A discussion of other proteases, effects of compensation and the significance of lon1 for plant physiology should be added to the text of the article.
- Figure 1 (original). What is the reason for the nonspecific staining? There are three stripes next to each other, how did the authors understand that the stripe in the middle is GFP-ATG8? Can these results be trusted? Why do the authors think it is possible to evaluate autophagy not only by GFP-ATG8, but also by GFP?
Response: Thank you for your inquiry. The nonspecific staining observed can be attributed to the low expression of ATG8A driven by the native promoter. In our autophagic flux assay, we employed pATG8A:GFP-ATG8A construct. Unlike the 35S/UBQ10 promoters known for robust and constitutive expression, resulting in higher gene expression levels, native promoters typically yield weaker expression. The relatively low expression level necessitates longer exposure times to ensure proper visualization of results, leading to nonspecific staining (as exemplified in the following western blot). For our positive control, Pro35S:GFP-ATG8A, both GFP-ATG8A and GFP were clearly detected within a short exposure time (10s). However, due to the low expression level driven by the native promoter in pATG8A:GFP-ATG8A, target bands were not initially visible. Upon extending the exposure time to 300s, nonspecific bands increased, but the target bands of pATG8A:GFP-ATG8A eventually appeared, matching the size of Pro35S:GFP-ATG8A (highlighted in red). Despite the presence of nonspecific bands, the target bands of GFP-ATG8A and GFP remain distinctly visible and of the correct size. Notably, our experiments were replicated three times biologically, yielding consistent results. We have provided the western blotting results for tree biological replicates in the modified Figure1. У
I don't see biological replicates. I see 4 tracks corresponding to 4 different types of plants. Even on the right side of the figure I see 4 bands, not 2. How did the authors understand that dedicated bands are GFP-ATG8 and GFP? What is the molecular weight (kDa) of these proteins? Did the authors use other controls or methods to confirm these proteins (may be GFP fluorescence)? Why do the authors think it is possible to evaluate autophagy not only by GFP-ATG8, but also by GFP? Why does Fig. 1Ð’ show not only GFP-ATG8, but also GFP?
- Fig.6 A, B. Why in the article, not in the supplement, are the results shown not for the three mutants separately, but in total? What is the significance of such a presentation of results?
Response: Thank you for your valuable feedback. Since the result is consistent across all three mutants, the analysis remains unchanged whether they are examined individually or collectively. Presenting them together offers the advantage of clearly identifying the genes commonly downregulated across the mutants, thereby elucidating those involved in brassinosteroid (BR)
biosynthesis and homeostasis. The shared downregulation of BR biosynthesis genes aligns with the abnormal seed development observed in all three mutant lines, as depicted in Figure 5. Hence, to enhance the logical flow, we have chosen to present this result in Figure 6.
The explanation of such data presentation should be added to the text of the article.
The conclusion should be rewritten so that it is consistent and reflects all the main findings. Now these are two sentences without context and the third sentence - general conclusion.

Round 3
Reviewer 1 Report
Comments and Suggestions for Authors
I again thank the authors for their detailed responses and additional materials provided. The authors significantly improved the manuscript. The article may be accepted for publication.